# Gene Expression Signatures for Guiding Initial Therapy in ER+/HER2- Early Breast Cancer

**DOI:** 10.3390/cancers17091482

**Published:** 2025-04-28

**Authors:** Sara Marín-Liébana, Paula Llor, Lucía Serrano-García, María Leonor Fernández-Murga, Ana Comes-Raga, Dolores Torregrosa, José Manuel Pérez-García, Javier Cortés, Antonio Llombart-Cussac

**Affiliations:** 1Medical Oncology Department, Hospital Arnau de Vilanova, 46015 Valencia, Spain; 2Doctoral School, Catholic University of Valencia San Vicente Mártir, 46001 Valencia, Spain; 3Translational Oncology Group, Facultad de Ciencias de la Salud, Universidad Cardenal-Herrera-CEU, 46115 Alfara del Patriarca, Spain; 4Clinical Laboratory, Hospital Arnau de Vilanova, 46015 Valencia, Spain; 5Medical Oncology Department, Hospital Dr. Peset, 46017 Valencia, Spain; 6International Breast Cancer Center (IBCC), Pangaea Oncology, Quiron Group, 08017 Barcelona, Spain; 7Department of Medicine, Faculty of Biomedical and Health Sciences, Universidad Europea de Madrid, 28670 Madrid, Spain

**Keywords:** ER+/HER2- breast cancer, gene expression signature, neoadjuvant chemotherapy, neoadjuvant endocrine therapy, clinical trials

## Abstract

Gene expression signatures can guide the adjuvant treatment of estrogen receptor-positive/HER2-negative early breast cancer and are currently incorporated into international treatment guidelines. Genomic assays can identify patients with a good prognosis that may not require chemotherapy. The use of genomic platforms in breast cancer biopsies has been validated, allowing for their utilization in guiding treatment strategies for early breast cancer and determining whether to propose upfront surgery or systemic therapy. Here, we review the evidence supporting the most common gene expression signatures (MammaPrint, BluePrint, Oncotype Dx, PAM50, EndoPredict, and the Breast Cancer Index) to guide the treatment of ER+/HER2- early breast cancer from the outset.

## 1. Introduction

Luminal breast cancer accounts for approximately 70% of all invasive breast carcinomas. The standard adjuvant treatment for this subtype typically involves a combination of CT and ET, guided by the patient’s clinical and pathological features.

In recent decades, gene expression assays have been introduced into clinical practice, offering a more precise prognosis compared to using clinicopathological features alone. These assays also provide valuable information regarding the potential benefits of adjuvant treatments, aiding in the identification of patients who may benefit the most.

Neoadjuvant chemotherapy (NACT) and neoadjuvant endocrine therapy (NET) are increasingly used to downstage tumors, enable less invasive surgical interventions, or transform locally advanced, inoperable tumors into operable ones. NACT usually consists of 6–8 cycles of anthracycline-based and/or taxane-based CT. The treatment of choice in NET is aromatase inhibitors, and the duration is usually between 4 and 6 months, but it can be extended in the case of a response. Achieving a pCR or residual cancer burden (RCB)-0 or RCB-I has been associated with improved prognosis, especially in aggressive tumor subtypes [1]. Furthermore, the neoadjuvant response could serve as a valuable tool for tailoring subsequent adjuvant treatments [2,3].

Given the role of gene expression signatures in optimizing adjuvant therapies and the expanding application of neoadjuvant treatments, their potential utility in guiding therapy decisions upfront appears promising. Here, we review the evidence supporting the use of the most commonly available gene expression signatures in the initial therapeutic decision-making: MP, BP, Oncotype DX, PAM50 (Prosigna), EP, and the BCI.

## 2. Clinical Utility of Gene Expression Signatures in the Adjuvant Setting

The use of gene expression signatures is included in the international treatment guidelines to support and optimize decision-making in adjuvant systemic therapy for ER+/HER2- EBC patients. These genomic tools help identify patients for whom CT is beneficial and those with a low risk of recurrence and excellent prognosis who could avoid undergoing CT.

MP, a 70-gene signature, classifies ER+/HER2- EBC patients into high-risk or low-risk groups for distant metastasis. In the MINDACT trial, MP identified high-clinical-pathological risk patients with a low genomic risk in which CT offered a 1.5% benefit for 5-year distant metastasis-free survival (DMFS), with an 8-year DMFS rate of 89.4% (CI 95%: 86.8–91.5) [4,5]. MP could also identify ultra-low-risk patients with an excellent prognosis even without adjuvant systemic therapy with a 20-year breast cancer-specific survival rate of 94% [6]. More recently, MP has been integrated into the implementation of BP (an 80-gene signature), which complements the MP results with intrinsic subtyping. BP was developed using 200 breast cancer patient specimens and was validated in four independent cohorts (*n* = 784), and was later demonstrated to be a predictor of CT response [7,8].

The 21-gene signature Oncotype Dx was prospectively validated in the TAILORx trial with node-negative (N0) patients [9,10]. In this study, postmenopausal patients with a Recurrence Score (RS) of 11–25 showed no added benefit from CT and ET, with a 9-year overall survival of 93.9% and 93.8% for the ET and CT + ET arms, respectively. In the subgroup aged ≤ 50 years, those with an RS of at least 16 benefited from adjuvant CT, and the magnitude of the benefit increased with increasing RS. Subsequently, Oncotype Dx was validated in the RxPONDER study among N1 (1–3 positive nodes) patients and showed no CT benefit in postmenopausal N1 patients with an RS of 0–25, whereas premenopausal women who received chemoendocrine therapy had better outcomes than those who received ET alone [11].

The Prosigna assay is based on the expression of fifty cancer genes (PAM50) and eight reference genes, utilizing the nCounter^®^ Analysis System from Nanostring Inc. (Seattle, WA, USA). The test provides a risk of recurrence (ROR) score based on gene expression data and tumor size, and also assigns an intrinsic breast cancer subtype similar to the one initially described by Perou et al. [12]. It was retrospectively validated in the ABSCG-8 trial, which included postmenopausal N0 and node-positive women who received adjuvant ET exclusively. The results showed that those with a low ROR had a 10-year risk of recurrence of <3.5% [13]. Using the TransATAC sample collection, which consists of the samples from the tamoxifen and anastrozole arms of the ATAC trial, it was demonstrated that the ROR score provides more prognostic information in N0 endocrine-treated patients than the Oncotype Dx RS [14]. The ongoing OPTIMA study is addressing the value of ROR for lymph node-positive, ER+/HER2- pre- and postmenopausal patients (ISRCTN42400492).

Other gene expression assays include EP and the BCI. EP analyzes eleven genes (including three reference genes) from ribonucleic acid (RNA) using Reverse-Transcription Polymerase Chain Reaction (RT-PCR) and calculates a prognostic score to classify patients into low- and high-risk groups. Two retrospective analyses of ABCSG studies found that EP provides additional prognostic information on the risk of distant recurrence independent of clinical-pathological parameters (ABCSG-6: *p* = 0.010; ABCSG-8: *p* < 0.001). Additionally, EP was combined with lymph node status and tumor size to generate an overall risk score, known as EPclin. The BCI is a combination of two gene assays: the expression ratio of homeobox B13 (HOXB13) to interleukin 17B receptor (IL17BR) (H:I ratio) and the Molecular Grade Index, which classifies patients as low, intermediate, or high risk. It was retrospectively validated in N0 postmenopausal women treated with adjuvant tamoxifen or placebo from the Stockholm study and a multi-institutional cohort. For both cohorts, consistent high-risk BCI scores were a significant prognostic factor beyond standard clinicopathologic factors for early and late recurrence [15]. The BCI was also a predictor of late recurrence and was useful in identifying candidates for extended ET [16,17,18,19]. To our knowledge, there are no ongoing prospective randomized trials on EP or the BCI.

Using the TransATAC sample collection, the reason for the differences in the estimation of the risk of distant recurrence provided by Oncotype Dx, Prosigna, EP, and the BCI was investigated. There were moderate to strong correlations among the four molecular scores except for RS vs. ROR and RS vs. BCI. These differences were explained by the fact that Oncotype DX RS was primarily driven by estrogen-related features while ROR, EP, and the BCI were dominated by proliferative features [14].

When analyzing the subgroup of premenopausal patients in the TAILORx study, a benefit of chemoendocrine therapy was observed in patients aged ≤ 50 years with an RS of 16–25 [9]. For patients with an RS of 16–20, the benefit for the distant recurrence rate was 1.6% at 9 years, whereas for those with an RS of 21–25, the benefit increased to 6.5% at 9 years. In the RxPONDER trial, which included patients with an RS ≤ 25, among the premenopausal women, the 5-year invasive disease-free survival was 89.0% with ET and 93.9% with chemoendocrine therapy (hazard ratio: 0.60; 95% CI: 0.43–0.83; *p* = 0.002) [11]. In the updated MINDACT analysis published by Piccart, the exploratory analysis showed a benefit of CT in patients aged ≤ 50 years with high clinical risk and low genomic risk, with an absolute difference in DMFS of 5.0% (SE: 2.8; 95% CI: −0.5 to 10.4) [5]. These findings suggest that the benefit of CT in premenopausal women may be related to CT-induced amenorrhea, which could potentially be achieved with ovarian function suppression (OFS). Ongoing trials, such as the OPTIMA trial and the OFSET study (NCT05879926), which include OFS as part of the ET in all premenopausal women, may clarify this question.

## 3. Is Gene Expression Analysis on a Biopsy Comparable to That on a Surgical Specimen?

Studies on genomic platforms that were initially conducted in the adjuvant setting have typically been performed on surgical resection specimens (SRSs). However, there are situations where performing the genomic study on a core needle biopsy (CNB) is valuable, such as cases where surgical excisions provide insufficient tissue for the test or when it is necessary to decide upfront on the treatment approach (surgical or neoadjuvant) [20]. To enable the use of these genomic platforms in initial decision-making, it is important to establish the correlation between results obtained from CNBs and SRSs.

A small retrospective MP study analyzed 13 patients diagnosed with breast cancer to assess the feasibility of performing a genomic study on CNBs, with success achieved in >80% of cases [21]. Subsequently, Crozier et al. reported a high concordance between CNB and SRS MP results (90.9%, k = 0.817) and BP results (98.3%) [22].

Several studies have been conducted to validate the correlation between Oncotype-DX CNB and SRS results. Initially, a study of 24 patients with EBC reported a correlation of 0.83 (95% CI: 0.64–0.92) between CNB and SRS results [23]. Jakubowski et al. analyzed 919,701 samples, of which, 13% were biopsies and 87% were surgical excisions. They found that the distribution of RSs was similar between CNB and SRS samples [24]. Similarly, Peng Qi et al. confirmed in 50 EBC patients that the RS results from paired CNB and SRS samples showed a 72% correlation based on TAILORx cutoffs [25]. Furthermore, in a study of 80 EBC patients using two publicly available gene expression datasets, Orozco et al. reported a high correlation between the RS of core and surgical samples, with concordance rates of 91% and 82% in the different datasets [26].

Regarding the PAM50-based Prosigna assay, Prat et al. evaluated the concordance in subtype and correlation in ROR between CNBs and their corresponding SRSs in 30 paired samples. The correlation between the Prosigna ROR scores of paired CNBs and SRSs was high (r ≥ 0.90) [27].

Müller et. al. compared the performance of the RNA-based EP multigene test on CNBs and SRSs from 40 breast cancer patients. They concluded that the EP scores of paired samples were highly correlated (Pearson r = 0.92), with an excellent concordance in the classification into low or high risk of metastasis (overall agreement: 95%) [28].

## 4. Does Neoadjuvant Treatment Impact Gene Expression Profiles?

In luminal subtypes, the low pCR rates after NACT or NET jeopardize its value as a prognostic factor. The RCB index, which takes into account bidimensional measurements of the residual tumor bed, invasive tumor cellularity, and nodal disease burden, has been validated as a stronger predictor of long-term outcomes than pCRs [29,30]. Changes in Ki67 levels from baseline after 2 weeks of NET were shown to be prognostic and are accepted as a guide for adjuvant decisions [31]. Several authors have reported that the post-NACT Ki67 level is also an independent predictor in patients with luminal EBC that fail to achieve a pCR [32,33,34,35].

The application of genomic platforms to guide neoadjuvant treatment has facilitated the assessment of the prognostic significance of molecular changes. Oncotype Dx has proven effective in predicting clinical responses to neoadjuvant therapy [36]. Ueno et al. demonstrated that the combination of pretreatment and post-treatment RSs is a reliable predictor of disease-free survival (DFS) (*p* = 0.002) and that it can distinguish between early and mid/late recurrences [37]. Moreover, Ciriaco et al. investigated variations in RS values using paired tumor samples collected before and after NACT. Their analysis revealed a significant post-NACT decrease in RSs, with an average reduction of 4.6 points. Additionally, 53.3% of patients with a baseline RS > 25 experienced a conversion to a low RS category after NAC [38]. More recently, the Dxcartes trial explored the biological and clinical effects of letrozole combined with palbociclib as a neoadjuvant therapy for patients with luminal EBC and an initial RS ≥ 18. Among the high-risk RS patients, 54.5% achieved molecular downstaging (defined as RS ≤ 25 or a pCR). These downstaging results suggest that the letrozole and palbociclib combination could be effective for aggressive luminal tumors [39].

Regarding PAM50 (Prosigna), the baseline score identified suitable patients for NET (Luminal A and Luminal B), excluding those with uncommon non-luminal intrinsically ET-resistant tumors (HER2-enriched or basal-like) [40]. Moreover, the ROR score after NET provides independent prognostic information beyond the PEPI score. The Neoendo study demonstrated that NACT and NET lowered the ROR plus a proliferation index (ROR-P) score. Indeed, a decrease in ROR-P and a change in the intrinsic subtype to the normal-like subtype suggest an improvement in event-free survival (EFS) [41]. These results were confirmed through the analysis of the genomic and molecular changes induced by the neoadjuvant treatment in ER+/HER2-low and ER+/HER2-0 tumors [42]. In the PROMIX trial, PAM50 was used to evaluate biopsies at baseline, after cycle 2 of NACT, and at the time of surgery. The intrinsic subtype after cycle 2 was not a significant predictor of the likelihood of achieving a pCR (PAM50, *p* = 0.5), but a statistically significant association was noted between the subtype after cycle 2 and both EFS (Log-rank *p* = 0.003) and overall survival (OS) (Log-rank *p* = 0.006) [43]. The CORALEEN study compared ribociclib combined with ET as a neoadjuvant strategy for NACT. The study observed a downstaging to a low risk of recurrence (low ROR) at surgery in both treatment groups [44].

There are data on EP changes from a study that assessed the efficacy of neoadjuvant palbociclib therapy in twenty postmenopausal women by evaluating its impact on cell cycle arrest and changes in EP scores before and after treatment. The mean EP score decreased with Palbociclib + ET, from 6.87 before treatment to 5.25 after treatment (*p* < 0.0001). The Preoperative Endocrine Prognostic Index (PEPI) is useful after NET as a predictor of survival, except in the case of neoadjuvant CDK4/6 inhibitors, where a usual washout period before surgery often leads to a Ki67 level rebound. Therefore, EPclin was compared with the PEPI score. The results suggested that EPclin might be a better parameter for estimating prognosis after neoadjuvant CDK4/6 inhibitor therapy [45].

Using MP and BP to identify transcriptional changes in EBC after NACT was evaluated by Chung et al. in a cohort of 128 women. Matched samples were used to analyze transcriptional differences between tumors that changed from MP high risk to low risk and those that remained high risk after NACT. Patients who remained MP high risk post-NACT had worse outcomes, indicating the need for additional therapy [46].

## 5. Gene Expression Signatures in the Neoadjuvant Setting

### 5.1. MammaPrint and BluePrint

Retrospective studies were conducted to evaluate the utility of MP and BP in predicting chemosensitivity in the neoadjuvant setting (Table 1) [47,48]. Ma et al. published the largest retrospective study that demonstrated that tumors classified as high risk by MP were more likely to achieve a pCR (*p* = 0.006) [49]. Additionally, no significant differences were found when comparing MP high-risk to RS ≥ 26 patients (*p* = 0.12). However, the MP high-risk group was less likely to achieve a pCR compared to those with an Oncotype RS ≥ 31 (*p* = 0.03).

As mentioned before, the correlation between the pCR rate, long-term outcomes, and the MP and BP molecular subtyping versus clinical subtyping was explored in 437 patients from four NACT trials [47]. The pCR rate differed substantially among the BP molecular subgroups: it was 6% in Luminal A-type, 10% in Luminal B-type, 47% in HER2-type, and 37% in basal-type patients. In the Luminal A-type group (*n* = 90, including 7 HER2-positive patients and 8 TNBC patients according to Immunohistochemistry/Fluorescence In Situ Hybridization (IHC/FISH)), the 5-year DMFS rate was 93% and the pCR rate provided no prognostic information, suggesting that these patients may not benefit from CT.

In the prospective NBRST, the MP index was highly associated with the likelihood of a pCR (*p* < 0.001) [50]. The BP assay reclassified 15% of clinical luminal tumors as the basal type (ER+/Basal), with a pCR rate similar to that of TNBC/basal tumors (34% vs. 38%; *p* = 0.52) and higher than that of Luminal A (2%; *p* < 0.001) and Luminal B (6%; *p* < 0.001) tumors [51]. A sub-analysis of patients treated with NET revealed similar partial response rates between Luminal A and Luminal B patients. However, the 5-year outcomes were significantly worse in those with Luminal B tumors (DMFS: 75.6% vs. 91.1%, *p* = 0.047) [52].

In the NBREaST study, Göker et al. showed that MP/BP could reclassify 9% of tumors, enhancing the accuracy of treatment response predictions. Patients with Luminal A-type tumors showed the poorest response to NACT but maintained the most favorable 5-year DMFS rate (91.4%), suggesting that these patients can forego NACT without compromising the outcomes [53]. The MINT trial reported that a greater proportion of patients with MP high-risk tumors treated with NACT underwent nodal downstaging compared to those with low-risk tumors (50.4% vs. 17.4%; *p* = 0.007) [54].

The Korean PLATO trial included patients who were initially ineligible for BCS and personalized neoadjuvant treatment based on their MP results. NACT and NET were administered to the high-risk and low-risk cases, respectively. Up to 27% of patients were MP low risk and received NET. The BCS rate was 58.9% overall (63.6% for NACT and 45.8% for NET). This approach could increase BCS rates and reduce unnecessary CT [55].

In the I-SPY1 study, a new median score cut-point of −0.154 was introduced to stratify patients into MP High 1 (MP1) or MP High 2 (MP2) groups, with MP1 defined as ≥−0.154 and MP2 defined as <0.154, to improve chemo-sensitivity predictions. A total of 138 patients were treated with NACT and a higher pCR percentage was observed for MP2 patients (*p* = 0.038) [56]. These findings were confirmed in the adaptative I-SPY2 trial, which included 379 ER+/HER2- EBC patients; the results showed a higher pCR in MP2 vs. MP1 patients (31% versus 11%, *p* = 1.1 × 10^−5^). The MP2, BP basal type subgroup that were ImPrint-positive were more likely to achieve a pCR to NACT with or without targeted agents or immunotherapy [57].

### 5.2. Oncotype Dx

The initial evidence for the use of Oncotype Dx in the neoadjuvant setting comes from small retrospective studies investigating the association between Oncotype DX RS and pCR rates [58,59]. A meta-analysis confirmed that the pCR rate was significantly higher in patients with a high RS compared to those with a low–intermediate RS (11% vs. 1%, respectively), suggesting that the RS may be useful in selecting patients for NACT [60].

In another retrospective study, Pardo et al. (2021) analyzed the association between axillary pCRs and RSs in 158 women diagnosed with clinical N1/N2, ER+/HER2 invasive ductal carcinoma who received NACT. Of the 23 patients who had an axillary pCR, 11 (47.8%) had a high-risk RS, 6 (26.1%) had an intermediate RS, and 6 (26.1%) had a low RS [61].

A multicenter study of premenopausal women aged <40 years examined the association between RSs and response to NACT. Of these women, 50% had an RS > 25. The pCR rate in tumors with an RS > 25 was 21% (8/38) versus just 5% in tumors with an RS < 25 (2/38) (*p*  =  0.09). In the multivariable analysis, only the RS result was significantly associated with a pCR (OR: 1.07; 95% CI: 1.01–1.12; *p*  =  0.01) [62].

In 2021, Morales Murillo et al. published the results of another prospective single-center study that enrolled 122 patients with ER+/HER2- invasive breast cancer who were candidates for NACT based on tumor size and/or biological criteria. The primary endpoint was to assess the distribution of pretreatment biopsy RS results. The treatment was assigned according to the RS results: patients with an RS < 11 underwent surgery and postmenopausal patients with an RS > 25 or premenopausal patients with an RS > 20 received NACT. The secondary endpoint was to assess RS outcomes based on the pCR [63]. It was found that 17.2% of patients had a low risk (RS < 11), 47.5% had an intermediate risk (RS of 11–25), and 35.2% had a high risk (RS > 20 or 25 depending on menopausal status). The PCR after NACT in the high-risk group was 12% and 47%, respectively. However, it was 0% in the intermediate- and low-risk groups.

Using a similar study design, Gasol et al. found a correlation between pCRs and RSs (Pearson: *p* = 0.001). Patients with an RS > 30 had an overall pCR rate of 37% compared to 14% in patients with an RS < 30. In addition, patients with a high Oncotype Dx RS > 30 who did not achieve a pCR had the worst outcomes, with a 20% relapse rate at the 32-month follow-up [64].

More recently, the KARMA Dx study by Llombart et al. sought to determine whether 21-gene test results influence physician treatment recommendations in the adjuvant and neoadjuvant settings. Cohort C of this study included 31 patients that were ER+/HER2- cT2 or cT3, cN0, and Ki67 level ≥ 30%. They compared the treatment recommendations made before and after the RS testing. After obtaining the RS results, the physicians changed their treatment recommendations, omitting CT for 76% of patients in cohort C, 80% of patients in the RS 0–17 group, and 83% in the intermediate RS group (18–30), who received ET instead of CT. Two out of three patients in the RS 31–100 group received NACT [65].

The prospective–retrospective transNEOS study validated the 21-gene test as a predictor of the clinical response to NET with letrozole in postmenopausal patients. The response rates were significantly higher in the low-RS group than in the intermediate- and high-RS groups (55% vs. 42% vs. 22%, respectively). Similarly, a higher likelihood of disease progression was associated with a high RS (17%) compared to an intermediate (4%) or low RS (<1%). Regarding conservative surgery, patients in the RS < 18 group were more likely to convert from BCS non-candidates to BCS candidates after NET than patients in the RS > 30 group (*p* = 0.010) [66].

Similar results were found in a systematic review and meta-analysis of eight prospective studies including 691 patients with ER+/HER2- breast cancer who had completed NET. The patients with a low- or intermediate-risk RS from their core biopsy were four times more likely to respond to NET than those with a high-risk RS [67].

Other studies have shown that intermediate-risk RSs are associated with responses to both NET and NACT, suggesting that Oncotype Dx is a useful tool for de-escalating CT without compromising outcomes [68,69]. Taken together, all these studies confirm that low-RS patients are unlikely to respond well to CT but are more likely to respond to NET and higher RS values predict a greater likelihood of a good response to CT.

In recent years, the addition of cyclin-dependent kinase 4 and 6 inhibitors to ET in the neoadjuvant setting has been investigated, in combination with Oncotype DX, in two prospective trials [68,69].

The SAFIA trial was a prospective, multicenter, double-blind, neoadjuvant phase III trial involving 354 pre-/peri- or postmenopausal patients with operable stage II or IIIA luminal breast cancer. RS testing was performed centrally on biopsies to identify patients eligible for NET (RS < 31). The patients started with fulvestrant for induction (±goserelin) and those who responded were randomized to fulvestrant + palbociclib or fulvestrant + placebo groups. Overall, 93.4% of the patients showed hormone responsiveness and there were no significant differences in RSs between the groups (97% for RS 0–10, 93% for RS 11–25, and 95% for RS 26–30). The addition of palbociclib to fulvestrant had no effect on the response rate compared to the fulvestrant alone arm (*p* = 0.14) [70].

In 2024, the Dxcartes study evaluated changes in RSs induced by letrozole and palbociclib as a neoadjuvant therapy in patients with ER+/HER2- EBC and a baseline RS ≥ 18 who were separated into two cohorts: cohort A (RS 18–25) and cohort B (RS 26–100). The primary endpoint was the proportion of patients who maintained or achieved an RS < 25 or a pCR at surgery. In the high-risk RS group, molecular downstaging (defined as an RS < 25 or a pCR at surgery) was achieved in 54.5% of cases (*p* < 0.01). In cohort A, 68.8% achieved an RS < 25 but no patient achieved a pCR and thus failing to reach the primary endpoint (*p* = 0.982). These results suggest that this combination may be effective in aggressive luminal tumors with an RS ≥ 18, particularly those with an RS of 26–100. However, long-term survival data are needed to determine the clinical significance [39].

### 5.3. PAM50 (Prosigna)

The Chemo-Endocrine Score (CES) was developed by Prat et al. to predict endocrine and CT sensitivity. It is based on the Correlation Coefficients (CCs) of the Luminal A and basal-like centroids of the PAM50 classification algorithm (CES=CC of Luminal A–CC of basal-like centroids). The CES’s predictive ability was evaluated in four independent neoadjuvant datasets and four adjuvant datasets. Based on the CES, the tumors were classified as more endocrine sensitive (CES-E), more CT sensitive (CES-C), or uncertain (CES-U) and it was independently associated with the response to NACT or NET. The CES might be clinically useful in ER+/HER2- patients who are PAM50 ROR intermediate, as the CES result was found to be significantly associated with distant relapse-free survival (RFS) in these patients [71].

The ACOSOG Z1031 study, which compared the neoadjuvants exemestane, letrozole, and anastrozole, analyzed the PAM50-based intrinsic subtype as a secondary endpoint. PAM50 identified 3.3% of the non-luminal intrinsically endocrine-resistant tumors and detected Luminal A and Luminal B tumors with high NET responsiveness. The occurrence of a PEPI score of 0 after NET was higher in the Luminal A group than in the Luminal B group (27.1% vs. 10.7%; *p* = 0.004) [40].

Dunbier et al. analyzed the correlations between PAM50 and Ki67 changes after two weeks of NET with anastrozole. The PAM50 intrinsic Luminal A and Luminal B tumors exhibited similar deceases in Ki67 levels after treatment, but the residual Ki67 levels remained higher in the Luminal B group, suggesting that Luminal A and B tumors may benefit similarly from AI, but patients with Luminal B tumors have poorer long-term outcomes [72]. López Velasco et al. studied the correlation between PAM50 and biomarkers of the response to NET. They showed that tumors with a low risk of recurrence based on subtype (ROR-S)/P at diagnosis exhibited low Ki67 levels after NET, in contrast to those with a high ROR-S/P (both *p* < 0.0001). Additionally, tumors that showed a strong decrease in Ki67 levels after NET had a lower ROR-S/P at diagnosis compared to those with either an increase or no change in Ki67 levels (*p* = 0.0041 and *p* = 0.0071, respectively) [73]. In the ALTERNATE trial, it was observed that PAM50 analysis (ROR-P and intrinsic subtype) on pre-NET biopsies could predict ET resistance, defined as an early on-NET (week 4) Ki67 level >10%, as well as ET sensitivity, as indicated by the modified Preoperative Endocrine Prognostic Index (mPEPI-0) or a pCR at surgery after NET. Furthermore, Luminal A or low ROR-P cases were the least likely to have a Ki67 level > 10% at week 4 (13.7%, *p* < 0.0001; 10%, *p* < 0.0001) and the most likely to achieve mPEPI-0 or a pCR (26.5%, *p* < 0.0001; 30.6%, *p* < 0.0001) [74].

Regarding the PAM50 analysis of patients undergoing NACT, Prat et al. were the first to report an association between intrinsic subgroups and NACT responses in ER+/HER2– breast cancer. Their findings showed that non-luminal tumors (basal-like and HER2-enriched) had higher pCR rates than luminal tumors (Luminal A and B) (30.0% vs. 8.9%; adjusted OR = 4.20; 95% CI: 2.220–7.942). Moreover, the ROR-P score influenced the outcomes, with a 5-year distant RFS rate of 90.2% (95% CI: 82.5–98.6%) in the low-risk ROR-P group. Among the patients who did not achieve a pCR, both the intrinsic subtype and the ROR-P were significantly associated [75]. In another study, Prat et al. analyzed 180 independent CNB samples from ER+/HER2- patients who were treated with NACT. Both the ROR (*p* = 0.047) and intrinsic subtype were significant predictors of the response to NACT. A categorical analysis of intrinsic subtypes using a logistic regression model showed that Luminal A tumors had a significantly lower likelihood of responding to NACT compared to other intrinsic subtypes (odds ratio for Luminal A vs. non-Luminal A = 0.341, *p* = 0.037) [27].

Ohara et al. evaluated the value of PAM50 and IHC in predicting NACT responses in 124 patients with ER-positive breast cancer. In this study, the intrinsic Luminal A tumor subtype was a significant (*p*  =  0.031) predictor of no pCRs, independent of other clinicopathological parameters, including Ki67 levels. On the other hand, Luminal A identified by IHC was not significantly predictive of pCRs [76].

The prospective I-SPY1 trial studied intrinsic subtypes using PAM50 in addition to MP. They observed that the pCR rates after NACT were lower in the Luminal A intrinsic subgroup (3%) compared to the Luminal B (16%) HER2-enriched (50%) or basal-like (33%) groups. The low-risk signatures identified patients with a good prognosis even without achieving a pCR, as the Luminal A intrinsic subgroup, which did not achieve pCRs, had a 94% 3-year RFS (relapse-free survival) rate [77].

In a cohort of 458 patients treated with NACT, Jensen et al. evaluated the predictive value of adding a multigene profile (CIT256 and PAM50) to the IHC profile. They observed a clear distinction in the pCR rate for patients with the CIT256 non-luminal vs. luminal subtype, (with an estimated odds ratio of 9.78 (95% confidence interval (CI): 2.60; 36.8, *p* < 0.001) and a similar odds ratio estimate for PAM50 (non-luminal vs. luminal: 8.82; 95% CI: 2.60; 29.0, *p* < 0.001). These results support the use of multigene profiles to classify tumors into luminal tumors, with a pCR likelihood of 3%, and non-luminal tumors, with a pCR probability of over 20% [78] to guide the neoadjuvant strategy.

Studies have also analyzed targeted therapies, such as the LETLOB phase II trial, which randomized 92 postmenopausal women with ER+/HER2- breast cancer to receive neoadjuvant letrozole + lapatinib or letrozole + placebo for 6 months. The PAM50 analysis showed that patients with non-luminal tumors (HER2-enriched or basal-like) showed reduced sensitivity to endocrine-based treatment with a lower objective response rate compared to those with luminal tumors (47% vs. 78%, *p* = 0.031) and a significantly higher Ki67 level at surgery after treatment (post-treatment median KI67 level: non-luminal (10%) vs. luminal (7%), *p* = 0.004) [79].

Several studies analyzed using PAM50 to predict the response to CDK4/6 inhibitors. NeoPal showed that real-time selection based on PAM50 intrinsic subtypes is feasible for a neoadjuvant trial [80]. The CORALEEN study used PAM50 to select Luminal B patients and compared ribociclib plus letrozole to NACT. At surgery, around 46% of both the ribociclib + letrozole and NACT groups had a low ROR score. This suggests that some patients with high-risk luminal EBC could achieve molecular downstaging with a CDK4/6 inhibitor and NET [44].

A study was conducted to evaluate the impact of the PAM50 assay on neoadjuvant treatment decision-making in patients with early-stage ER+/HER2- breast cancer, as well as patients’ confidence in their treatment plan. The results showed a change in treatment decisions in 28% of cases, while 45% of physicians reported an increase in confidence regarding the treatment plan after PAM50 testing and 62% of patients experienced a reduction in anxiety about their treatment [81].

### 5.4. EndoPredict

There are three retrospective studies that evaluated the predictive value of EP (12-gene molecular score) for the response to neoadjuvant treatment (NACT and NET). Bertucci et al. assessed the predictive value of EP for a pCR to NACT in 553 patients with ER+/HER2- EBC. The EP classification correlated with a pCR rate of 7% in the low-risk group and 17% in the high-risk group (*p* < 0.001). The 5-year DFS rate was significantly higher in the low-risk patients (88%) compared to the high-risk patients (73%) (*p* = 0.015), indicating that the greater chemosensitivity in high-risk tumors does not compensate for their increased risk of recurrence and lower sensitivity to ET [82].

Dubsky et al. evaluated the ability of EP to predict the response to NACT or NET in a retrospective–prospective translational study using samples from ER+/HER2- EBC patients within the ABCSG-34 trial. They analyzed 134 patients who received NACT (93% with high-risk EP) and 83 who received NET (53% with low-risk EP). The 12-gene molecular score (MS) significantly predicted the treatment response to both NACT (AUC: 0.736; 95% CI: 0.63–0.84) and NET (AUC: 0.726; 95% CI: 0.60– 0.85). In the NET group, a greater proportion of patients with low-risk disease (27.3%) achieved RCB 0-I compared to those with high-risk disease (7.7%). However, the response to NACT was better among high-risk patients, with 24.6% attaining RCB 0-I, whereas none (0%) of the low-risk EP patients exhibited this response [83].

A study by Soliman et al. compared the ability of the 12-gene MS and the 21-gene RS to predict pCRs to NACT in ER+/HER2- EBC patients using expression data from 764 patients across six publicly available RNA expression microarray datasets obtained from the Gene Expression Omnibus database. When analyzed separately, both scores were significant predictors of pCRs (12-gene MS: *p* = 9.4 × 10⁻⁵; 21-gene RS: *p* = 0.0041). However, when analyzed together in the same model, EP remained significant (*p* = 0.0079), whereas Oncotype did not (*p* = 0.79). Furthermore, the 12-gene MS proved to be a superior predictor compared to the 21-gene RS in the combined analysis, regardless of the proliferation group threshold [84].

### 5.5. Breast Cancer Index

In 2011, Mathieu et al. published a French retrospective study that included 150 samples from ER+/HER2- EBC patients treated with NACT. They explored the ability of the BCI to predict pCRs and BCS rates. BCI classified 42% of patients as low risk, 35% as intermediate risk, and 23% as high risk. In a multivariate analysis that incorporated clinicopathologic parameters, the BCI showed a significant association with pCRs. When categorized into risk groups, the odds ratio for the high-risk versus low-risk group was 34 (*p* = 0.0055). Regarding BCS, the BCI risk groups remained significantly associated with being BCS candidates, with an OR of 5.78 for the high- vs. low-risk group (*p* < 0.0001 and *p* = 0.0022, respectively). This was the first study that demonstrated the ability of the BCI to predict eligibility for BCS and sensitivity to NACT [85].

Spring et al. investigated the ability of BCI to predict pCRs in ER+ EBC patients treated with NACT by analyzing 185 pre-treatment core biopsies. The BCI was significantly associated with pCRs, with no pCRs observed in the low-risk group (*p* = 0.0003) and it was the only significant predictor of pCRs in the multivariate analysis that included clinicopathological factors (OR = 14.0; 95% CI: 1.19–263.5; *p* = 0.05) [86].

### 5.6. Other Gene Expression Signatures

TheraPrint is a microarray platform that analyzes the expression of 125 genes that were selected for their potential relevance to prognosis and therapeutic responses. De Snoo et al. evaluated its predictive and/or prognostic value in two patient cohorts treated with NACT. Several overlapping genes between the two datasets were found to be significantly differentially expressed between responders and non-responders. The study identified TheraPrint genes with a statistically significant correlation between expression levels and response to NACT [87]. Beitsch et al. correlated the chemosensitivity results (measured using pCRs) in the NBRST to the TheraPrint results. This study identified 13 genes with a statistically significant correlation between expression and response to NACT, suggesting that TheraPrint could provide predictive information about the response to NACT [88].

**Table 1 cancers-17-01482-t001:** Main prospective trials of the five commercially available gene expression signatures in the neoadjuvant setting.

Signature	Trial	Study Design	Primary Endpoint Results
MammaPrint BluePrint	NBRST Whitworth et al. [50,51] Pellicane et al. [52]	Prospective Registry	pCR to NACT: 11% Clinical response to NET: 62%
	NBREASTII Göker et al. [53]	Prospective	pCR to NACT Luminal A: 2% Luminal B: 12% Basal-like: 50% Partial response to NET Luminal A: 76.5% Luminal B: 80%
	PLATO Han et al. [55]	Prospective Phase II	BCS conversion rate: 70% (95% CI: 59.4–79.2%)
	MINT Blumencranz et al. [54]	Prospective Phase IV	Nodal downstaging rate: 45.2% Association with MP: *p* = 0.007 Association with BP: *p* < 0.001
	I-SPY1 Wolf et al. [56]	Prospective Phase II	pCR MP1: 15% MP2: 33% (*p* = 0.038)
	I-SPY2 Huppert et al. [57]	Prospective Phase II	pCR MP1: 11% MP2: 31% (*p* = 1.1 × 10^−5^)
PAM50 (Prosigna)	ALTERNATE Ma et al. [74]	Prospective Phase III	ESDR = pCR or mPEPI 0 Anastrozole: 18.7% Fulvestrant: 22.8% A + F: 20.5%
	ACOSOG Z1031 Ellis et al. [40]	Prospective Phase II	cRR Exemestane: 62.4% Letrozole: 74.8% Anastrozole: 69.1%
	I-SPY1 Esserman et al. [77]	Prospective Phase II	RFS Luminal A: 94% Luminal B: 79% HER2-enriched: 90% Basal-like: 58% Normal-like: 100% pCR Luminal A: 3% Luminal B: 16% HER2-enriched: 50% Basal-like: 33% Normal-like: 33%
	LETLOB Griguolo et al. [79]	Prospective Phase II	ORR Non-luminal: 47% Luminal: 78% (*p* = 0.031)
	CORALEEN Prat et al. [44]	Prospective Phase II	ROR low-risk at surgery Ribociclib + Let: 46.9% NCT: 46.1%
Oncotype Dx	Morales et al. [63]	Prospective	RS distribution in NACT candidates RS < 11: 17.2% RS 11–25: 47.5% RS > 25: 35.2%
	Gasol et al. [64]	Prospective	pCR RS < 30: 14% RS > 30: 37% Recurrence rate RS < 30: 12% RS > 30: 15%
	KARMA Dx Llombart-Cussac et al. [65]	Prospective	Impact of RS on treatment plan Change to omit CT: 67% (95% CI: 60–73%)
	SAFIA Alsaleh et al. [70]	Prospective	pCR ET +- Palbociclib: 7 vs. 2% (*p* = 0.14)
	Dxcartes Guerrero-Zotano et al. [39]	Prospective	RS ≤ 25 at surgery or pCR RS 18–25: 68.8/0% RS 26–100: 54.5/3%
EndoPredict	Dubsky et al. [83]	Prospective Phase II	RCB 0–1 NET EP low risk: 12 pts EP high risk: 3 pts NCT EP low risk: 0 pts EP high risk: 33 pts

BCS: breast-conserving surgery; BP: BluePrint; cRR: clinical response rate; DRFS: distant recurrence-free survival; MP: MammaPrint; mPEPI: modified preoperative endocrine prognostic index; NACT: neoadjuvant chemotherapy; NET: neoadjuvant endocrine therapy; pCR: pathological complete response; RCB: residual cancer burden; RFS: relapse-free survival; ROR: risk of recurrence; pts: patients; A: anastrozole; F: fulvestrant; ESDR: endocrine-sensitive disease rate; let: letrozole; ORR: objective response rate.

## 6. New Insights and Current Clinical Trials

The validation of genetic platforms for selecting neoadjuvant treatments in luminal breast cancer using clinical trials continues to evolve (Table 2). The NeoTAILOR study aims to utilize a novel biomarker-driven approach (PAM50, OncotypeDx, and Ki67 levels after 4 weeks of NET) to guide neoadjuvant treatment selection. The study is classifying ER+/HER2- EBC patients into intrinsic molecular subtypes based on PAM50 (non-Luminal A vs. Luminal A) or high vs. low risk based on RSs. The low-risk patients will receive NET for 6 months and the high-risk patients will receive NET alone or with CT depending on whether they exhibited early endocrine sensitivity, defined as a Ki67 level < 10% after 28 days on anastrozole. The primary endpoint is the objective response rate, measured using breast Magnetic Resonance Imaging, in the endocrine-sensitive group after 6 months of NET (NCT05837455).

The “Neoadjuvant Dose-dense Chemotherapy for HR+/HER2- Breast Cancer Patients With High Proliferation Index” study uses a high proliferation index score (Ki67 > 30%) to select high-risk ER+/HER-2- patients as candidates for neoadjuvant treatment with dose-dense paclitaxel followed by epirubicin cyclophosphamide. Moreover, MB/BP are being explored as biomarkers to identify which patients benefit more from dose-dense NACT. The RCB is being used as a treatment outcome (NCT05728268).

PROOFS is a Real-World Data (RWD) registry that collects data from pre-/perimenopausal patients on the use of ovarian function suppression, the effect of secondary amenorrhea (CT-induced or not), and the ET treatments administered. The correlation between the MP risk index and patient outcomes will be assessed under real-world conditions (NCT05792150) [89].

The “PersonaLized neoAdjuvant Strategy ER Positive and HER2 Negative Breast Cancer TO Increase BCS Rate” trial proposes using MP for risk assessment to guide neoadjuvant treatment in patients who are not candidates for BCS. The objective is to evaluate the BCS rates after neoadjuvant therapy with NACT (ACx4 followed by Tx4) in MP high-risk patients and NET in MP low-risk patients (NCT03900637) [55]. In the surgery scenario, the NACAGEP study uses MP to estimate the need for CT and thus administer a neoadjuvant strategy, with the intention of de-escalating surgery in high-risk patients (lumpectomy instead of mastectomy and a lymph node-sparing procedure instead of a full axillary lymph node dissection) and improving quality of life after surgery (NCT05666258).

Gene signatures are being used to select patients who are candidates to receive other agents in addition to conventional CT. An example is immune checkpoint inhibitors (ICIs). The phase III trial “Adding an Immunotherapy Drug, MEDI4736 (Durvalumab), to the Usual Chemotherapy Treatment (Paclitaxel, Cyclophosphamide, and Doxorubicin) for Stage II-III Breast Cancer” compares EFS rates and response rates (RRs) between patients classified as MP ultra-high treated with standard NACT alone or combined with the anti-PD-L1 treatment MEDI4736 (Durvalumab) (NCT06058377). The VALENTINE study randomized high-risk patients (ki67 ≥ 20% and/or high genomic risk) into three arms (NACT, Patritumab–Deruxtecan (Her3-Dxd) + ET, and Her3-Dxd) to evaluate the pCR at surgery as the primary endpoint (NCT05569811) [90].

CDK4/6 inhibitors are another therapeutic strategy for high-risk ER+/HER-2- patients. The RIBOLARIS trial (NCT05296746) evaluated whether CT can be omitted in patients initially classified as high risk based on clinicopathologic and/or genomic criteria but achieved pathological downstaging or a low-ROR after six months of treatment with ribociclib and letrozole, which was then continued as adjuvant therapy. The initial genomic risk could be assessed using Oncotype Dx, MP, Prosigna, or EP; however, the genomic risk at the time of surgery could only be determined by PAM50. The PREDIX LumB trial also selected patients using IHC or, preferably, genomic profiling using PAM50 signature (any Luminal B, Luminal A with node metastases, and/or ≤40 years old). They were randomized to receive neoadjuvant treatment, weekly paclitaxel or palbociclib with ET for 12 weeks, and then the treatment was switched (crossover). The primary objective is the radiological objective response after 12 weeks of treatment (NCT02603679). The study “Multigene Risk Score Combined with Ki-67 Dynamic Assessment in Stratified Neoadjuvant Endocrine Therapy Treatment With or Without CDK4/6 inhibitors in ER+/HER2- Breast Cancer” evaluates the efficacy of the neoadjuvant CDk4/6 inhibitor dalpiciclib in high-risk EPclin and non-ET responders (defined as no decrease in Ki67 levels after two weeks of Letrozole). They are also exploring predictive biomarkers for sensitivity to CDK4/6 inhibitor therapy (NCT06650748).

**Table 2 cancers-17-01482-t002:** Ongoing gene expression signature trials in the neoadjuvant setting.

Trial Identifier	Phase	Patients	Gene Signature	Treatment Strategy	Reference
NCT05837455 (NeoTAILOR)	II	ER+/HER2- postmenopausal, stage II/III	PAM50, Oncotype DX	ET ± standard CT	No results posted
NCT05728268	II	ER+/HER2-, stage IIB-IIIC, Ki67 level > 30%	MammaPrint, BluePrint	Dose-dense nab-paclitaxel followed by EC	No results posted
NCT05792150 (PROOFS)	Observational	ER+/HER2- premenopausal and perimenopausal	MammaPrint	ET ± OFS CT vs. no CT	Fischer et al. [89]
NCT03900637 (PLATO)	II	ER+/HER2-, stage I–IIIA, non-BCS candidates	MammaPrint	CT vs. ET	Han et al. [55]
NCT05666258 (NACAGEP)	Observational	ER+/HER2-, stage cT1-3 cN1	MammaPrint	CT and surgery de-escalation	No results posted
NCT06058377	III	ER+/HER2-, stage II–III Men or women	MammaPrint	NACT ± MEDI4736	No results posted
NCT05569811 (VALENTINE)	II	ER+/HER2- primary operable, Ki67 level > 20% and/or PAM50 high risk Men or women	PAM50	CT, HER3-DXd ± ET	Oliveira et al. [90]
NCT05296746 (RIBOLARIS)	II	ER+/HER2-, stage II, grade 2–3, Ki67 level ≥ 20% Men and women	PAM50, Oncotype Dx, MammaPrint, EndoPredict	Ribociclib + ET or CT	No results posted
NCT02603679 (PREDIX LumB)	II	ER+/HER2-, >20 mm with/without lymph node metastasis Men or women	PAM50	Standard CT vs. Palbociclib + ET	No results posted
NCT06650748	II	ER+/HER2-, Ki67 level > 20%, stage T2N1	EndoPredict	ET ± Dalpiciclib	No results posted

BCS: breast cancer surgery; CT: chemotherapy; ER: estrogen receptor; ET: endocrine therapy; HER2: human epidermal growth factor receptor-2; Her3-Dxd: Patritumab–Deruxtecan; ICIs: immune checkpoint inhibitors; OFS: ovarian function suppression; RR: response rate; RWD: Real-World Data.

## 7. Conclusions

Gene expression signatures are widely used in ER+/HER2- EBC in the adjuvant context, which is supported by prospective phase III studies and recommended by international clinical guidelines to guide adjuvant treatment. With the rise of neoadjuvant strategies, the use of these genomic assays before treatment has been proposed to predict the response to NET or NACT, but the available data are limited.

It has been observed that the most commonly used gene expression signatures show a high correlation with CNB and SRS results. Moreover, the molecular changes in SRS samples induced by NET and NACT resulted in a reduced risk compared to those in CNBs. The information provided by this molecular downstaging can be added to the changes in Ki67 expression after 2–4 weeks of NET, as well as the RCB and the PEPI scores, which are currently used to guide post-surgery treatments.

Most studies have evaluated the capacity of gene expression signatures to predict the response to NACT or NET. There is a consensus on the strong correlation between high genomic risk and pCR rates following NACT. In addition, patients with low/intermediate-risk tumors have a higher probability of responding to NET. The studies are also consistent with the very favorable outcome of Luminal A/low-risk tumors independent of their poorer response to NACT, with RFS rates greater than 90% at 3–5 years.

Studies incorporating targeted therapies, mainly CDK4/6 inhibitors, used genomic assays to initially classify patients. In some studies, such as CORALEEN, the post-treatment genomic result was also used to assess the molecular response. The results suggested that some Luminal B tumors may achieve enough pathological downstaging with CDK4/6 inhibitors and ET to not require CT in the future.

Most studies exploring the use of genomic platforms to guide neoadjuvant treatment were retrospective and had a small sample size. More consistent phase II and III trials are ongoing, which will provide more information to support the use of gene assays to guide the therapeutic plan before undergoing surgery.

## Data Availability

This manuscript does not include new research data.

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
