# Peer review of "Gene Expression Signatures for Guiding Initial Therapy in ER+/HER2- Early Breast Cancer"

_cancers, 2025, doi:10.3390/cancers17091482_

Round 1

Reviewer 1 Report

Comments and Suggestions for Authors

Your article “Gene Expression Signatures for Guiding Initial Therapy in ER+/HER2- Early Breast Cancer” discusses several gene expression signatures in ER+/HER2- breast cancer. Would it be helpful to mention other gene expression panels such as TheraPrint as a predictor of response to neoadjuvant chemotherapy? In sentence 52 in introduction please correct the spelling of chemotherapy.

Author Response

Reviewer #1: Your article “Gene Expression Signatures for Guiding Initial Therapy in ER+/HER2- Early Breast Cancer” discusses several gene expression signatures in ER+/HER2- breast cancer.

Comment 1: Would it be helpful to mention other gene expression panels such as TheraPrint as a predictor of response to neoadjuvant chemotherapy?

Response: Thank you very much for your suggestions. We have included section “5.6. Other Gene Expression Signatures”, where we mention TheraPrint.

“TheraPrint is a microarray platform that analyzes the expression of 125 genes selected for their potential relevance to prognosis and therapeutic response. De Snoo et al. evaluated its predictive and/or prognostic value in two patient cohorts treated with NACT. Several overlapping genes between the two datasets were found to be significantly differentially expressed between responders and non-responders. The study identified TheraPrint genes with a statistically significant correlation between expression levels and response to NACT [87]. Beitsch et al. correlated chemosensitivity (measured by pCR) in the NBRST to TheraPrint results. This study identified 13 genes with statistically significant correlation between expression and response to NACT, suggesting TheraPrint could provide predictive information about the response to NACT [88].”

Comment 2: In sentence 52 in introduction please correct the spelling of chemotherapy.

We have corrected the spelling error in the word “chemotherapy”.

Reviewer 2 Report

Comments and Suggestions for Authors

The manuscript “Gene Expression Signatures for Guiding Initial Therapy in ER+/HER2- Early Breast Cancer” is a typical example of a very good effort to bring order into somewhat chaotic world of various approaches (used in commercialization efforts) of gene expression testing It thoroughly reviews the validation of various panels for selecting neoadjuvant treatments in breast cancer, with references to clinical trials. Although not a systematic review and somewhat procedural in nature, it is undeniably useful. It is possible to recommend accepting the manuscript after minor improvements.

In Table 2 give citations (numbers) instead of doi.

Better give an example with citation for this text: “neoadjuvant response could serve as a valuable tool for tailoring subsequent adjuvant treatments”.

Please clarify “those in whom could be avoided”.

Better put inside parentheses this text: “ in the pCR rate for patients grouped in the CIT256 Non-luminal vs. Luminal subtypes, with an estimated odds ratio of 9.78 (95% confidence interval (CI) 2.60;36.8), p< 0.001, and a similar odds ratio estimate for PAM50 Non-luminal vs. Luminal 8.82 (95%CI 2.60;29.0), p < 0.001”.

Please refine the sentences containing these text fragments:

  1. “as investigated the reason for the differences in the estimation of de risk of distant recurrence”
  2. “hNACT y NET “
  3. “clyclin-dependent kinase “
  4. “The patients' confidence in their treatment plan was evident.”
  5. “Asociación con M”

Finally, at least one graphic figure should greatly improve the presentatin of this work.

Comments on the Quality of English Language

There are numerous imperfections perhaps due to a fast transcription to English from Spanish or related language, and these must be carefully fixed.

Author Response

Reviewer #2: The manuscript “Gene Expression Signatures for Guiding Initial Therapy in ER+/HER2- Early Breast Cancer” is a typical example of a very good effort to bring order into somewhat chaotic world of various approaches (used in commercialization efforts) of gene expression testing It thoroughly reviews the validation of various panels for selecting neoadjuvant treatments in breast cancer, with references to clinical trials. Although not a systematic review and somewhat procedural in nature, it is undeniably useful. It is possible to recommend accepting the manuscript after minor improvements.

Comment 1: In Table 2 give citations (numbers) instead of doi.

Response: Thank you very much for your suggestions and for the positive evaluation. We have modified the references in Table 2, indicating the citation instead of the DOI.

Comment 2: Better give an example with citation for this text: “neoadjuvant response could serve as a valuable tool for tailoring subsequent adjuvant treatments”.

Response: At this point, we refer to how adjuvant treatment is determined based on the response to neoadjuvant therapy, as in the case of capecitabine in the CREATE-X study and T-DM1 in the KATHERINE trial. We have added the citations for these studies.

Comment 3: Please clarify “those in whom could be avoided”.

Response: We have clarified in the manuscript that chemotherapy (CT) could be avoided in patients with a low risk of recurrence and excellent prognosis: “These genomic tools help identify patients for whom CT is beneficial and those with a low risk of recurrence and excellent prognosis who could avoid undergoing CT”.

Comment 4: Better put inside parentheses this text: “ in the pCR rate for patients grouped in the CIT256 Non-luminal vs. Luminal subtypes, with an estimated odds ratio of 9.78 (95% confidence interval (CI) 2.60;36.8), p< 0.001, and a similar odds ratio estimate for PAM50 Non-luminal vs. Luminal 8.82 (95%CI 2.60;29.0), p < 0.001”.

Response: We have put this text inside parentheses.

Comment 5: Please refine the sentences containing these text fragments:

“as investigated the reason for the differences in the estimation of de risk of distant recurrence”

“hNACT y NET “

“clyclin-dependent kinase “

“The patients' confidence in their treatment plan was evident.”

“Asociación con M”

Response: We have reviewed and revised the sentences you indicated making the necessary changes to correct the errors.

“In the TransATAC sample collection, the reason for the differences in the estimation of de risk of distant recurrence provided by Oncotype Dx, Prosgina, EP and BCI was investigated”.

“NACT and NET”

“cyclin-dependent kinase”

“as well as the patients' confidence in their treatment plan”.

“Association”

Comment 6: Finally, at least one graphic figure should greatly improve the presentating of this work.

Response: We appreciate your suggestion; however, we have been unable to find a way to include a figure as well for this work.

Comment 7: Comments on the Quality of English Language. There are numerous imperfections perhaps due to a fast transcription to English from Spanish or related language, and these must be carefully fixed

Response: In order to improve the quality of the English language, we have sent the manuscript to MDPI Author Services for English editing.

Reviewer 3 Report

Comments and Suggestions for Authors

This article summarizes the application and value of gene expression profiling in neoadjuvant therapy for early ER+/HER2- breast cancer. The summary is accurate, the terminology is standardized, and the logic is rigorous. Below are some minor suggestions for improvement:

P2L52: It would be better to provide a more detailed explanation of the definitions of neoadjuvant chemotherapy (NACT) and neoadjuvant endocrine therapy (NET).

The use of abbreviations in the article is inconsistent—some terms are abbreviated after being introduced, while their full forms are used again later.

There are several grammatical and spelling errors throughout the article. The authors should carefully proofread the entire text. For example, on P4L171, there is a spelling mistake in the word "performance." On P6L257, "suggesting" is misspelled

Author Response

Reviewer #3: This article summarizes the application and value of gene expression profiling in neoadjuvant therapy for early ER+/HER2- breast cancer. The summary is accurate, the terminology is standardized, and the logic is rigorous. Below are some minor suggestions for improvement:

Comment 1: P2L52: It would be better to provide a more detailed explanation of the definitions of neoadjuvant chemotherapy (NACT) and neoadjuvant endocrine therapy (NET).

Response: Thank you very much for your suggestions and for the positive evaluation. The revised text from the manuscript defining NACT and NET is as follows: “NACT usually consists of 6-8 cycles of anthracycline-based and/or taxane-based CT. The treatment of choice in NET is aromatase inhibitors, and the duration is usually between 4 and 6 months, but it can be extended in the case of a response”.

Comment 2: The use of abbreviations in the article is inconsistent—some terms are abbreviated after being introduced, while their full forms are used again later.

Response: We have reviewed the manuscript to ensure the correct use of abbreviations and to avoid using full forms once the abbreviation has been introduced. I have highlighted in yellow the modified abbreviations.

Comment 3: There are several grammatical and spelling errors throughout the article. The authors should carefully proofread the entire text. For example, on P4L171, there is a spelling mistake in the word "performance." On P6L257, "suggesting" is misspelled.

Response: We have reviewed the spelling and grammatical errors in the manuscript. Furthermore, we have submited it to MDPI Author Services for English editing.